# Most Earth-surface calcites precipitate out of isotopic equilibrium

M. Daëron[1], R.N. Drysdale[2,3], M. Peral[1], D. Huyghe[4,5,6], D. Blamart[1], T.B. Coplen[7], F. Lartaud[4] & G. Zanchetta[8]

Oxygen-isotope thermometry played a critical role in the rise of modern geochemistry and remains extensively used in (bio-)geoscience. Its theoretical foundations rest on the assumption that $^{18}O/^{16}O$ partitioning among water and carbonate minerals primarily reflects thermodynamic equilibrium. However, after decades of research, there is no consensus on the true equilibrium $^{18}O/^{16}O$ fractionation between calcite and water ($^{18}\alpha_{cc/w}$). Here, we constrain the equilibrium relations linking temperature, $^{18}\alpha_{cc/w}$, and clumped isotopes ($\Delta_{47}$) based on the composition of extremely slow-growing calcites from Devils Hole and Laghetto Basso (Corchia Cave). Equilibrium $^{18}\alpha_{cc/w}$ values are systematically ~1.5‰ greater than those in biogenic and synthetic calcite traditionally considered to approach oxygen-isotope equilibrium. We further demonstrate that subtle disequilibria also affect $\Delta_{47}$ in biogenic calcite. These observations provide evidence that most Earth-surface calcites fail to achieve isotopic equilibrium, highlighting the need to improve our quantitative understanding of non-equilibrium isotope fractionation effects instead of relying on phenomenological calibrations.

[1] Laboratoire des Sciences du Climat et de l'Environnement, LSCE/IPSL, CEA-CNRS-UVSQ, Université Paris-Saclay, Orme des Merisiers, F-91191 Gif-sur-Yvette Cedex, France. [2] School of Geography, The University of Melbourne, 221 Bouverie Street, Carlton, VIC 3053, Australia. [3] EDYTEM UMR CNRS 5204, Bâtiment "Pôle Montagne", Université Savoie Mont Blanc, 5 bd de la Mer Caspienne, F-73376 Le Bourget du Lac Cedex, France. [4] Laboratoire d'Ecogéochimie des Environnements Benthiques, LECOB, Sorbonne Université, CNRS, F-66650 Banyuls-sur-mer, France. [5] Géosciences Environnement Toulouse, UMR 5563 CNRS, UR 234 IRD, UM 97 UPS, Observatoire Midi-Pyrénées, CNES, 14 avenue Édouard Belin, 31400 Toulouse, France. [6] Centre de Géosciences, MINES ParisTech, PSL University, 35 rue St Honoré, 77305 Fontainebleau Cedex, France. [7] U.S. Geological Survey, 12201 Sunrise Valley Drive, Reston, VA 20192, USA. [8] Dipartimento di Scienze della Terra, Università di Pisa, Via S. Maria 53, 56126 Pisa, Italy. Correspondence and requests for materials should be addressed to M.D. (email: daeron@lsce.ipsl.fr)

Harold Urey's prediction[1], based on fundamental thermodynamic principles, that the isotopic composition of carbonate minerals must be strongly influenced by their crystallization temperature is the cornerstone of both stable-isotope geochemistry and paleoclimatology. $^{18}O/^{16}O$ abundance ratios in carbonates primarily reflect the temperature and oxygen-isotope composition of the water from which they precipitated, both of which vary in complex but generally understood ways under the influence of important environmental parameters, such as altitude, latitude, atmospheric circulation, greenhouse gas concentrations, global ice volume, and rainfall distribution. For 7 decades, this relationship has been extensively applied to the study of past climates[2], sedimentary and diagenetic processes in the Earth's crust[3], the global carbon cycle[4], biological mechanisms of calcification[5], ore deposits and petroleum geology[6], the petrogenesis of carbonatites[7], the ecology of marine and terrestrial species[8], and the early Solar System[9]. This versatility stems from the ubiquity of carbonate minerals in the geologic record and the biosphere, and from the ease with which dissolved inorganic carbon (DIC) species exchange oxygen isotopes with water, reaching chemical and isotopic equilibrium on short time scales[10].

Despite the historical and contemporary importance of oxygen-isotope geochemistry, there is still no consensus on the true equilibrium $^{18}O/^{16}O$ fractionation factors between carbonates and water[11–16]. Different groups of inorganic and biogenic carbonates appear to follow different fractionation laws, prompting the use of phenomenological calibrations believed to describe the behavior of various specific types of natural carbonates (e.g., refs. [17,18]). Although many of these calibrations display similar temperature sensitivities, with $\delta^{18}O$ values decreasing by 0.2‰ per K around 20 °C, certain types of carbonates, such as speleothems[19,20], corals[5,21], or coccoliths[22], are clearly influenced by additional parameters beyond precipitation temperature. These discrepancies most likely reflect isotopic disequilibrium related to poorly constrained kinetic/metabolic processes, consistent with the fact that Earth-surface carbonates generally precipitate rapidly from supersaturated solutions[23]. Our understanding of such non-equilibrium processes, however, is far from complete. The main unanswered questions concern which DIC species are directly involved in crystallization, the processes by which different reaction pathways may be favored or inhibited, and how to quantitatively describe nucleation effects, crystal-surface phases, or the role of amorphous calcite[24–26].

To isolate and understand these non-equilibrium processes, it is necessary to establish a baseline of equilibrium $^{18}O/^{16}O$ fractionation as a function of temperature. Both theory[23,27] and empirical results[28] suggest that full attainment of oxygen-isotope equilibrium might not be achievable in laboratory experiments, so that one must instead turn to natural minerals precipitated very slowly from slightly supersaturated environments. Suitable natural carbonates, however, remain exceedingly rare. It has been argued, based on its extremely slow growth and the long-term stability of its geochemical environment, that the subaqueous mammillary calcite coating the walls of the Devils Hole cave system (Nevada, USA) offers optimal conditions for equilibrium crystallization[16]. Devils Hole calcite of Holocene age, precipitated at ~33.7 °C, is known to yield significantly higher $\delta^{18}O$ values than those predicted from laboratory experiments and from many biogenic calcite calibration studies, suggesting that most natural carbonates are affected by non-equilibrium oxygen-isotope fractionations with magnitudes on the order of 1–2‰. Several recent theoretical models of kinetic fractionation[23,27,29] have postulated equilibrium fractionation factors anchored to the Devils Hole data, but relying on a single observation remains problematic, particularly when extrapolating to colder environments.

Here, we extend the isotopic equilibrium baseline to low temperatures based on another instance of extremely slow-growing calcite, originating from an unusual karstic environment. We find that this equilibrium baseline displays a slope (i.e., temperature sensitivity) indistinguishable from that for faster-growing calcite, with a constant oxygen-18 enrichment of ~1.5‰. We also compare the clumped-isotope ($\Delta_{47}$) compositions of these two slow-growing calcites to that of biogenic calcite produced by bivalves and foraminifera, and also observe subtle but resolvable $\Delta_{47}$ differences between "equilibrium" and biogenic calcite. We conclude that most calcites precipitating at the surface of the Earth fail to achieve complete isotopic equilibrium.

## Results

**Slow-growing calcite from Laghetto Basso.** The subaqueous calcite coating found at the bottom of Laghetto Basso, a small lake in Corchia Cave (Italy), provides an apparently continuous paleoclimate record of the last 960 ka[30]. In situ observations of pH and temperature spanning more than 10 years, along with numerous isotopic and elemental analyses of water samples (ref. [31] and Supplementary Table 1), demonstrate that modern pool water is thermally and chemically stable, with pH = 8.2 ± 0.1 and $T$ = 7.9 ± 0.2 °C (1 SD). Drip counting conducted over several hours in May 2017 suggests that the lake received 50–60 L per day during this period. Based on an estimated lake volume of 20 m$^3$, water residence time is expected to be on the order of 1 year, much longer than the ~33 h required for 99% isotopic equilibration between DIC and water. What's more, in contrast to most karstic environments of paleoclimatic interest, dripwater must percolate through the Corchia Cave system for long durations on the order of years to decades before reaching Laghetto Basso[31]. As a result, the subaqueous calcite precipitates from a solution which is already very close to chemical and isotopic equilibrium with host rocks and the local cave atmosphere[30,31].

Laghetto Basso calcite shares many other similarities with Devils Hole mammillary calcite, making it very likely that it was also precipitated in isotopic equilibrium. Both sites are characterized by low values of calcite saturation indices (0.18 ≤ log ($\Omega$) ≤ 0.30), very slow growth rates (≤ 0.8 μm/y), similar surface textures and crystal fabrics, and comparable solution ratios of [DIC]/[Ca2+] and [Mg2+]/[Ca2+][31,32]. In the context of the present study, the most significant difference between the two sites is the higher pH in Laghetto Basso (8.2 versus 7.4 at Devils Hole). Although pH is expected to influence $^{18}O/^{16}O$ fractionation between water and rapidly-precipitating calcite, this effect decreases with slower crystallization rates[27,33], and becomes negligible (≤ 0.05‰) at the very slow growth rates considered here.

**Oxygen-18 equilibrium.** The oxygen isotope compositions of Devils Hole and Laghetto Basso waters are known from earlier studies, with respective $\delta^{18}O_{VSMOW}$ values of −13.54 ± 0.05‰[16] and −7.39 ± 0.09‰ (refs. [20,34], Supplementary Table 1). We sampled calcite from the outer surface of coatings from both sites and measured their carbon and oxygen stable-isotope compositions (Table 1). Both samples yield calcite/water oxygen-18 fractionation factors ($^{18}\alpha_{cc/w}$) which are 1.5‰ greater than predicted by the experimental calibration of Kim and O'Neil[15] (Fig. 1), defining an equilibrium baseline (Eq. (1), with crystallization temperature $T$ in kelvin) whose slope is indistinguishable from that of the synthetic precipitates:

$$10^3 \ln\left(^{18}\alpha_{cc/w}\right) = 17.57 \times 10^3/T - 29.13 \qquad (1)$$

The regression uncertainties are best expressed by reformulating the above equation so that regression errors in its slope and

**Table 1 Crystallization conditions and stable-isotope compositions of water and calcite from Devils Hole and Laghetto Basso**

|  | Devils Hole | Laghetto Basso |
|---|---|---|
| Sample | DVH | LGB |
| Average pH | 7.4 | 8.2 |
| Ionic strength | $10.5 \times 10^{-3}$ | $5.2 \times 10^{-3}$ |
| Growth rate (mol m$^{-2}$ s$^{-1}$) | $1-8 \times 10^{-10}$ | $\sim 3 \times 10^{-10}$ |
| Temperature (°C ± 1 SD) | $33.7 \pm 0.2$ | $7.9 \pm 0.2$ |
| Water $\delta^{18}O_{VSMOW}$ (‰ ± 1SE) | $-13.54 \pm 0.05$ | $-7.39 \pm 0.09$ |
| Calcite $\delta^{13}C_{VPDB}$ (‰ ± 1SE) | $-1.95 \pm 0.01$ | $0.02 \pm 0.02$ |
| Calcite $\delta^{18}O_{VPDB}$ (‰ ± 1SE) | $-15.83 \pm 0.04$ | $-4.48 \pm 0.03$ |
| 1000 ln($^{18}\alpha_{cc/w}$) (±1SE) | $28.13 \pm 0.06$ | $33.38 \pm 0.10$ |
| $\Delta_{47}$ (‰ ± 1SE) | $0.6309 \pm 0.0041$ | $0.7247 \pm 0.0040$ |

Because of low supersaturation conditions and extremely slow growth rates, the composition of these two natural samples is very likely to record equilibrium values of $^{18}\alpha_{cc/w}$ and $\Delta_{47}$

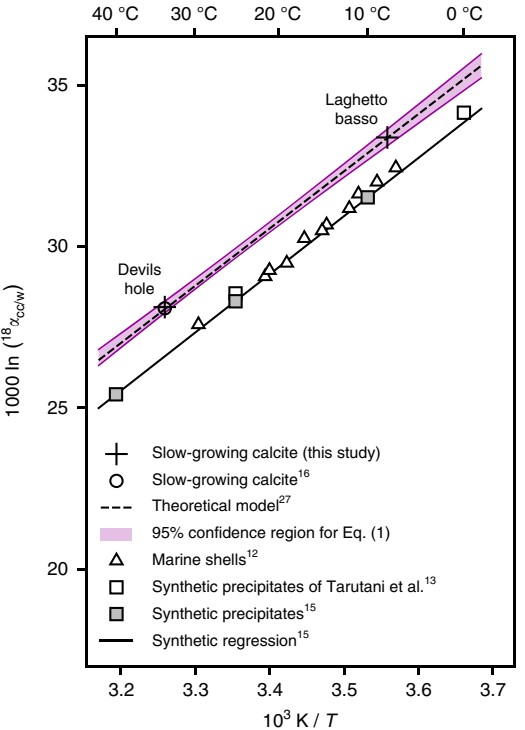

**Fig. 1** Equilibrium $^{18}O/^{16}O$ fractionation between calcite and water ($^{18}\alpha_{cc/w}$) as a function of crystallization temperature ($T$). The equilibrium baseline defined by slow-growing calcites from Devils Hole and Laghetto Basso (colored confidence region, Eq. (2)) is indistinguishable from the theoretical prediction of Watkins et al.[27] (dashed line), which is quasi-identical to the original prediction by Coplen[16]

intercept values are independent:

$$10^3 \ln(^{18}\alpha_{cc/w}) = A \times 10^3 (1/T - 1/T_0) + B$$
$$A = 17.57 \pm 0.43 \ (1 \ SE)$$
$$B = 29.89 \pm 0.06 \ (1 \ SE) \tag{2}$$
$$T_0 = 297.7 \ K$$

The temperature sensitivity of Eq. (1) is 0.20‰ per K at 20 °C, which is similar to that of equilibrium oxygen-18 fractionation between dissolved (bi)carbonate ions ($CO_3^{2-}$, $HCO_3^-$) and water (0.19‰ and 0.21‰ per K, respectively)[35]. Our findings are thus consistent with the hypothesis that the kinetic components of

$^{18}\alpha_{cc/w}$ vary primarily with pH, crystallization rate, and/or solution saturation, but remain relatively insensitive to temperature (at least within the range of typical Earth-surface conditions), as postulated in several theoretical models[27,29].

**Clumped-isotope disequilibrium in biogenic calcites.** As a complementary characterization of isotopic equilibrium, we also measured the clumped-isotope composition ($\Delta_{47}$) of these two calcite samples (Table 1). Clumped isotopes describe statistical anomalies in the abundance of isotopologues with multiple rare isotopes, such as $(^{13}C^{18}O^{16}O_2)^{2-}$ [36]. In the same way that carbonate $\delta^{18}O$ values potentially record equilibrium oxygen-isotope fractionation factors between the mineral and aqueous phases, $\Delta_{47}$ values of calcite are expected to reflect temperature-dependent isotopic equilibrium constants within the mineral phase[37], providing a complementary but independent isotopic thermometer.

The Devils Hole–Laghetto Basso calibration for equilibrium values of $\Delta_{47}$ in calcite at Earth-surface temperatures (Fig. 2a) is described by the following equation:

$$\Delta_{47} = 46.0 \times 10^3/T^2 + 0.1423 \tag{3}$$

Again, reformulating Eq. (3) so that regression errors in its slope and intercept values are independent yields:

$$\Delta_{47} = A \times 10^3 (1/T^2 - 1/T_0^2) + B$$
$$A = 46.0 \pm 2.8 \ (1 \ SE)$$
$$B = 0.6786 \pm 0.0029 \ (1 \ SE) \tag{4}$$
$$T_0 = 292.9 \ K$$

The slope of this regression is statistically indistinguishable from those obtained by several recent $\Delta_{47}$ calibration studies[28,38–40]. However, precise comparisons between clumped-isotope measurements performed in different laboratories remain challenging due to several methodological issues[41,42]. For instance, earlier $\Delta_{47}$ measurements of Devils Hole calcite[43,44] are not directly comparable to the values reported here because they are anchored to $CO_2$ standards instead of the carbonate standards used in our study. To circumvent this problem, we compare our equilibrium observations to the clumped-isotope compositions of planktonic and benthic foraminifera collected from marine sediment core-tops[45] and of modern calcitic bivalves from environments with minimal seasonal variability, all of which were analyzed in a single laboratory, following identical analytical procedures, using the same set of carbonate standards, within a limited time frame (10 months).

Laghetto Basso calcite yields a slightly lower $\Delta_{47}$ value than the biogenic samples formed at similar temperatures, but this difference arguably remains within analytical uncertainties. By contrast, the clumped-isotope composition of Devils Hole calcite plots $17 \pm 5$ ppm (1SE) below the extrapolated foraminifer regression line, and $27 \pm 8$ ppm below the bivalve line. It is notable that Devils Hole calcite precipitates from waters with a significantly lower pH than most biogenic carbonates. For example, several foraminiferal species are known to actively elevate pH at calcification sites by at least 0.5 units above typical seawater pH values of 8.2[46–48]. However, pH is only expected to influence $\Delta_{47}$ in fast-growing carbonates[33,44,49,50]. Thus, if the biogenic carbonates analyzed here had achieved clumped-isotope equilibrium, they should not display large $\Delta_{47}$ departures from the DVH-LGB baseline regardless of pH.

One possible interpretation of these results is that biogenic samples formed at low temperatures achieve quasi-equilibrium clumped-isotope compositions, but warmer samples do not.

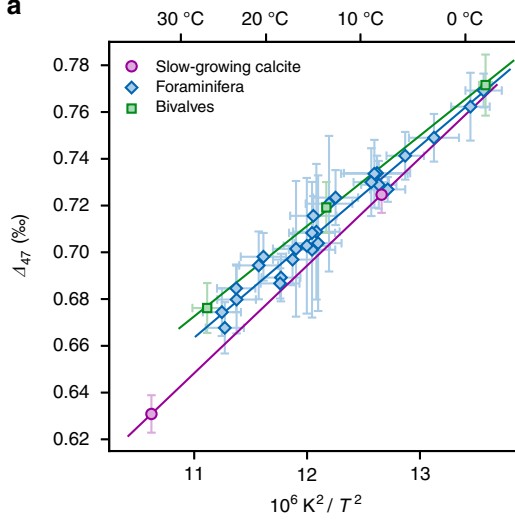

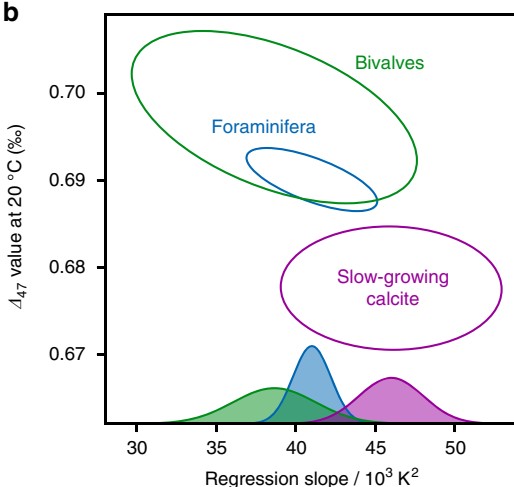

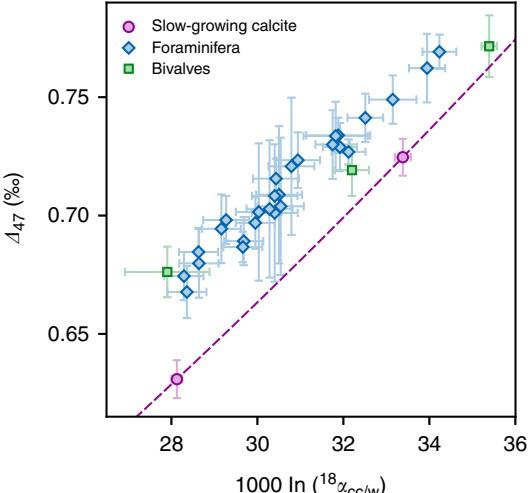

**Fig. 3** Equilibrium versus biogenic calcites in $^{18}\alpha_{cc/w} - \Delta_{47}$ space. Combining the $^{18}O/^{16}O$ and $\Delta_{47}$ thermometers requires some constraints on water $\delta^{18}O$ values, but makes it possible to test whether carbonates precipitated in isotopic equilibrium even if crystallization temperature is unknown. Dashed line corresponds to the equilibrium baseline defined by Eqs. (1) and (3). Error bars represent 95% confidence intervals

Our ability to jointly define equilibrium values for the two independent isotopic thermometers opens up interesting new possibilities. For instance, combining $\Delta_{47}$ and $^{18}\alpha_{cc/w}$ observations clearly exposes large isotopic differences between our biogenic carbonates and the slow-growing calcites, without requiring any assumptions on their crystallization temperatures (Fig. 3). We anticipate that this kind of combined observations will be most useful in studies such as those of diagenetic carbonates, where temperatures remain poorly constrained but where the oxygen-isotope composition of parent waters may be estimated from independent methods (e.g., fluid inclusions[51]).

## Discussion

Our findings demonstrate that mammillary calcite from Devils Hole is not an anomalous outlier, but rather that natural calcites formed at crystallization rates much slower than those achieved so far in laboratory experiments are systematically enriched in oxygen-18 relative to carbonates precipitating more rapidly from equilibrated DIC solutions. This observation offers support to theoretical models in which oxygen-18 fractionation between DIC and calcite ($^{18}\alpha_{cc/DIC}$) varies between an equilibrium limit and a kinetic limit, respectively corresponding to low versus high values of crystallization rate, saturation index, and ionic strength[23,27,29]. The fact that the slope of the equilibrium regression line in Fig. 1 is indistinguishable from that of Kim and O'Neil[15] or from that of equilibrium fractionation between dissolved (bi)carbonate ions and water[35] implies that both the equilibrium limit and the kinetic limit of $^{18}\alpha_{cc/DIC}$ do not vary strongly with temperature.

An important prediction of these theoretical models is that virtually all biogenic and most inorganic calcites precipitating at the surface of the Earth crystallize too rapidly to achieve DIC-calcite equilibrium. This conclusion is not invalidated by the fact that some rapidly-precipitating inorganic carbonates such as speleothems[18] or travertines[52] often display higher $\delta^{18}O$ values than predicted by Kim and O'Neil[15], because this observation may be simply explained by isotopic disequilibrium between DIC and water due to Rayleigh fractionation of the DIC pool under conditions of rapid $CO_2$ degassing[53]. Carbonates formed close to isotopic equilibrium are only expected to be found in

**Fig. 2** Three calibrations of clumped isotopes in carbonates ($\Delta_{47}$) as a function of crystallization temperature $T$. **a** Observed relations between $\Delta_{47}$ and $T$ in slow-growing calcite from Devils Hole and Laghetto Basso, in modern calcitic bivalves (Supplementary Table 2), and in foraminifera from sedimentary core-tops (data from Peral et al.[45]), all of which were analyzed in the same laboratory over a short period of time. Error bars represent 95% confidence intervals. Solid regression lines take into account analytical errors in $\Delta_{47}$ as well as uncertainties on crystallization temperature. **b** Comparison of the 95% confidence regions of regression slopes and 20 °C intercept values for slow-growing and biogenic calcite. Both of the biogenic regression lines differ significantly ($p \leq 10^{-3}$) from the equilibrium baseline defined by slow-growing calcites. Colored bell-shaped curves represent the probability distributions of regression slopes

Alternatively, if we assume that clumped isotopes in biogenic samples and in slow-growing calcites are characterized by the same regression slope, the $\Delta_{47}$ values of foraminifera and bivalves are respectively $11 \pm 3$ and $17 \pm 5$ ppm (1SE) higher than predicted from the equilibrium baseline. More generally, despite statistically indistinguishable regression slopes (Fig. 2b), an analysis of covariance based on conservative estimates of analytical errors and temperature uncertainties indicates that the observed difference between slow-growing inorganic calcite and the biogenic samples is statistically significant ($p \leq 10^{-3}$). Contrary to the case of oxygen isotopes, these differences are not much larger than the current precision limits on $\Delta_{47}$ measurements, particularly when taking inter-laboratory discrepancies into account.

environments with very low supersaturation states, such as for example recrystallized carbonates from deep-sea sediments[54], or carbonates associated with low-temperature hydrothermal alteration of young oceanic crust[55]. Deeper away from the surface, isotopic equilibrium might be the rule rather than the exception for diagenetic or metamorphic carbonates formed at significantly warmer temperatures, where isotope exchange reaction rates are much faster.

The biogenic carbonates analyzed here yield $\Delta_{47}$ values 5–20 ppm higher than equilibrium, and it appears possible that the magnitude of clumped-isotope disequilibrium decreases at low calcification temperatures. It should be noted that "oxygen-18 equilibrium", referring to oxygen-isotope exchange between water and mineral phases, and "clumped-isotope equilibrium", referring to the internal distribution of isotopes within the mineral phase, are logically independent, i.e., neither implies the other, because ultimately they reflect different processes. It is still an open question whether the clumped-isotope signature of calcite is inherited from that of one or more DIC species, or whether it reflects partial or complete isotopic exchanges occurring in transitional phases such as amorphous calcite or crystal-surface phases[25,26,28,44]. By contrast, achieving oxygen-18 equilibrium between water and calcite requires establishing a series of intermediate equilibria: between water and DIC, then between DIC and calcite, either directly or through the intermediate phases mentioned above. Each of these exchange steps may fail to achieve equilibrium, which potentially manifests in very different ways. For example, rapid $CO_2$ degassing of DIC solutions is associated with kinetic isotope fractionation effects which strongly affect both $\delta^{18}O$ and $\Delta_{47}$[56], contrary to the disequilibrium observations reported here which only weakly affect the latter.

Our findings provide robust new evidence that the majority of calcites precipitated at the surface of the Earth achieve neither oxygen-18 nor clumped-isotope equilibrium, probably because most of them precipitate rapidly from supersaturated solutions. In most cases, kinetic components of $^{18}\alpha_{cc/DIC}$ typically decrease carbonate $\delta^{18}O$ values by 1–2‰, even in "well-behaved" biogenic carbonates where $^{18}\alpha_{cc/w}$ varies primarily with temperature. As noted by Watkins et al.[27], oxygen-isotope thermometry works reasonably well in spite of these strong kinetic effects because many types of natural carbonates precipitate under limited ranges of pH and growth rates. However, the observation that non-equilibrium oxygen-18 effects in coccolithophores have varied drastically at geologic time scales[22,57] offers a cautionary tale regarding the long-term applicability of modern calibrations for biogenic carbonates. Moving beyond phenomenological characterizations of oxygen-isotope and $\Delta_{47}$ thermometry calls for substantial improvements in our ability to model isotopic fluxes and fractionations in the water/DIC/carbonate system. In our view, the use of non-classical isotopic tracers, such as clumped isotopes and oxygen-17 anomalies ($\Delta^{17}O$), offers appealing new opportunities to test and improve these models.

## Methods

**Inorganic calcite samples.** Holocene Devils Hole calcite (sample DVH) was collected from the outer surface of sample DHC2-8, which was previously described by Winograd et al.[58] and Coplen[16]. After a 15-min ultrasonic bath treatment with reagent-grade methanol, we abraded the surface of DHC2-8 to a maximum depth of 100 μm using a programmable micro-mill at its slowest setting. Laghetto Basso calcite (sample LGB) was collected from the top of core CD3-12, located a few centimeters away from core CD3, which was described by Drysdale et al.[30]. Each half of CD3-12 was ultrasonically cleaned in deionized water to remove loose particles from the active growth surface, then air-dried at ambient temperature. Calcite was abraded from 15 discrete 1-cm$^2$ regions of its outer surface using a Dremel hand tool fitted with a diamond burr and a magnification lens. The depth of abrasion was estimated to be no more than 100 μm. Both DVH and LGB powders were then rinsed in methanol and dried at room temperature.

**Bivalve samples.** Three specimens of Antarctic scallop species *Adamussium colbecki* were collected at a water depth of 15 m near the Dumont d'Urville Antarctic station in January 2007 (66.658°S, 140.008°E). Seawater temperature, constrained by the ROSAME network (Réseau d'Observation Sub-Antarctique et Antarctique du niveau de la MEr), remains stable annually (mean $T = -1.8$ °C) except for a summer warming peak around $-0.5$ °C between January and March[59]. Seawater $\delta^{18}O_{VSMOW}$ value, estimated from the Global Seawater Oxygen-18 Database of Schmidt et al.[60], is $-0.26 \pm 0.06$‰.

Five live specimens of the deep-sea oyster species *Neopycnodonte cochlear* were collected in March 2010 from the Lacaze-Duthiers canyon (42.533°N, 3.453°E, Mediterranean Sea) at a depth of 270 m, about 20 km east of the coast. Mean annual temperature remains constant at $13.5 \pm 0.1$ °C[61]. Local $\delta^{18}O_{VSMOW}$ values vary seasonally between 0.23 and 0.93‰, with an average value of 0.70‰ (M. Sebilo, pers. comm.).

Four live *Saccostrea cucullata* oysters from the warm shallow waters of the Kenyan coast (Tiwi Beach, 4.239°S, 39.604°E) were collected in September 2005. Local seawater temperatures vary annually from 25.1 to 28.5 °C ($T = 26.8 \pm 0.9$ °C).

All bivalves were rinsed with deionized water and bathed in 5% $H_2O_2$ to remove organic matter. Subsampling of *N. cochlear* and *S. cucullata* targeted the hinge area of each specimen, potentially offering a complete record of life-long calcification. Approximately 15 mg calcite powder was collected from each hinge area using a Dremel hand tool fitted with a 0.2-mm bit. For *A. colbecki*, we selected a small piece of the shell and ground it manually in an agate mortar.

**Foraminifer samples.** Peral et al.[45] analyzed Late Holocene foraminifera collected from 13 marine sediment core-tops, comprising 9 planktonic and 2 benthic species. Calcification temperatures were estimated based on the gridded seawater $\delta^{18}O$ model of LeGrande and Schmidt[62], assuming the oxygen-18 fractionation law of Kim and O'Neil[15]. Note that the observed differences between the slow-growing inorganic calcites and the foraminifera only increase if calcification temperatures were derived instead from the oxygen-18 fractionation law of Shackleton[63].

**Traditional stable-isotope analyses.** Traditional stable-isotope analyses ($\delta^{13}C$, $\delta^{18}O$) of samples DVH and LGB were performed using a MultiCarb system coupled to an Isoprime 100 mass spectrometer in dual-inlet mode. International carbonate standards NBS 19 ($\delta^{13}C_{VPDB} = 1.95$‰; $\delta^{18}O_{VPDB} = -2.20$‰) and NBS 18 ($\delta^{13}C_{VPDB} = 5.01$‰; $\delta^{18}O_{VPDB} = -23.01$‰) were analyzed along with DVH and LGB. All samples and standards were analyzed six times, with each replicate analysis requiring about 150 μg of carbonate. Sample $\delta^{13}C$ and $\delta^{18}O$ values were computed directly from ion current ratios 45/44 and 46/44 using the IUPAC-recommended oxygen-17 correction parameters of Brand et al.[64]. As recommended by Coplen[65], final $\delta^{18}O_{VPDB}$ values are scaled to the nominal oxygen isotope compositions of NBS 19 and NBS 18. The overall external reproducibility (standard deviation) of these measurements were 0.02‰ for $\delta^{13}C_{VPDB}$ and 0.04‰ for $\delta^{18}O_{VPDB}$.

**Clumped-isotope analyses.** Clumped isotope measurements were performed according to previously described protocols[41,45]. Carbonate samples were converted to $CO_2$ by phosphoric acid reaction at 90 °C. After cryogenic removal of water, the evolved $CO_2$ was helium-flushed through a purification column packed with Porapak Q and held at $-20$ °C, then quantitatively recollected by cryogenic trapping and transferred into an Isoprime 100 dual-inlet mass spectrometer equipped with six Faraday collectors ($m/z$ 44–49). Pressure-dependent background current corrections were measured independently for each sample. Background-corrected ion current ratios were converted to $\delta^{13}C$, $\delta^{18}O$, and "raw" $\Delta_{47}$ as described by Daëron et al.[41], using the IUPAC oxygen-17 correction parameters[64]. The raw $\Delta_{47}$ values were converted to the "absolute" $\Delta_{47}$ reference frame defined by the "ETH" carbonate standards[42]. The overall external reproducibility (standard deviation) of $\Delta_{47}$ measurements for carbonate samples and standards is 15 ppm. Average $\Delta_{47}$ values are based on 22 replicate analyses (each) for samples DVH and LGB, 20 replicates for *N. cochlear*, 17 for *S. cucullata*, and 12 for *A. colbecki*. Full analytical errors are derived from the external reproducibility of carbonate standards ($N = 151$) and samples ($N = 93$) within each analytical session, and conservatively account for the uncertainties in raw $\Delta_{47}$ measurements as well as those associated with the conversion to the "absolute" $\Delta_{47}$ reference frame.

**Statistical methods.** Relationships between $\Delta_{47}$ and crystallization temperature are modeled using weighted orthogonal distance regressions of the form $\Delta_{47} = A/T^2 + B$ in order to account for errors in both variables. In all three regressions, root mean square weighted deviation (RMSWD) values are smaller than one, implying that analytical and observational errors are sufficient to explain the scatter in the mean observations.

Analysis of covariance (ANCOVA) was performed by first computing the probability for the null hypothesis that two independent regression lines have identical slopes. If the two slopes are statistically indistinguishable (at a given confidence level), observations from both data sets are jointly fit to a new model with two parallel lines. If the distance between these two lines is statistically indistinguishable from zero, the null hypothesis that the two data sets follow the same relationship between $\Delta_{47}$ and $T$ cannot be excluded.

## Data availability
The complete raw data and all associated code used in this study are available under a Creative Commons license at https://doi.org/10.5281/zenodo.1227428.

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

## Acknowledgements

We thank C. Spötl and three anonymous reviewers for useful feedback on the manuscript. M.D. is grateful to T. Kluge for his early encouragement to write up this study. The clumped-isotope facility at LSCE is part of PANOPLY (Plateforme Analytique Géosciences Paris-Saclay) and was supported by the following institutions: Région Ile-de-France; Direction des Sciences de la Matière du Commissariat à l'Energie Atomique; Institut National des Sciences de l'Univers, Centre National de la Recherche Scientifique; Universtité de Versailles/Saint-Quentin-en-Yvelines. M.D. thanks N. Smialkowski, F. Dewilde, and C. Waelbroeck (LSCE) for "bulk" $\delta^{13}C_{VPDB}$ and $\delta^{18}O_{VPDB}$ analyses on LGB and DVH. This work was partially supported by Australian Research Council Discovery Project Grants DP110102185 (to R.N.D., G.Z., and M.D.) and DP160102969 (to R.N.D. and G.Z.). R.N.D. was the grateful recipient of a 2017 Jean D'Alembert Fellowship awarded through the Université Paris-Saclay and with support of LSCE. M.P. thanks the Commissariat à l'Energie Atomique for supporting her through "Thèse Phare" PhD fellowship. M.D, D.H., and F.L. gratefully acknowledge the support of Agence Nationale de la Recherche project AMOR, led by Y. Donnadieu (CEREGE). The authors thank Yves-Marie Paulet and Laurent Chauvaud (LEMAR, UBO) for the collection of *A. colbecki* specimens, supported by the MACARBI project (IUEM). The authors are also grateful to the captain and crew of the RV Nereis II from the UMS Service at Sea of the OOB, for their help in acquiring *N. cochlear* shells, and to M. Sebilo (IEES, Sorbonne Université) for oxygen-18 analyses of water from the Lacaze-Duthiers canyon. This is LSCE contribution #6534.

## Author contributions

M.D. initiated the project aiming to determine baseline equilibrium $\Delta_{47}$ values based on natural slow-growing calcite. He set up the clumped-isotope facility at LSCE and oversaw the quality of all clumped-isotope measurements; performed most of the clumped-isotope analyses of DVH and LGB; designed the statistical analysis and figures presented here; wrote the present report with primary contributions from R.N.D., D.B. and T.B.C., and additional contributions from other co-authors. R.N.D. conducted preliminary work that originally recognized that Laghetto Basso calcite could be used to constrain oxygen-isotope equilibrium fractionation at low temperatures; helped conceive the research; compiled the water chemistry data from Laghetto Basso; prepared and subsampled the outer surface of CD3-12. M.P. selected sedimentary core-tops; picked, identified, and cleaned foraminifera; performed foraminiferal clumped-isotope analyses; compiled seawater composition estimates. D.H. selected and subsampled bivalve specimens; performed bivalve clumped-isotope analyses. D.B. co-developed the LSCE clumped-isotope facility and sampled the outer surface of DHC2-8; performed some of the clumped-isotope analyses of DVH and LGB. T.B.C. provided sample DHC2-8. F.L. helped select bivalve specimens; collected seawater samples from Lacaze-Duthiers canyon; compiled seawater composition estimates for *A. colbecki*. G.Z. oversaw the geochemical monitoring of Laghetto Basso and collected core CD3-12.
