## [Peer Review File · Nature Communications]

Reviewer #1 (Remarks to the Author):

This study presents new isotopic measurements of extremely slow-growing cave calcite from Laghetto Basso, complementing measurements from Devils Hole, Nevada. The Devils Hole calcite has been described as one of the few examples of calcite, lab-grown or natural, that has grown in isotopic equilibrium with its source water. Because of the high temperatures under which Devils Hole calcite was grown, the equilibrium fractionation relationship established there was of limited use at other sites.

The site Laghetto Basso is hydrologically similar to Devils Hole, but is significantly cooler. Combining clumped- and traditional-isotope measurements of both Devils Hole and Laghetto Basso calcite, the authors have described equilibrium oxygen-isotope and clumped-isotope temperature calibrations that span a broad range of environmental temperatures. These new calibrations are consistent with previously-reported theoretical equilibrium calibrations based off of the Devils Hole calcite measurements. The authors use these new calibrations to argue that the vast majority of calcite grows out of oxygen and clumped-isotope equilibrium with its source water.

This is a strong paper that adds further evidence that oxygen-isotope equilibrium in calcite is the exception rather than the rule. Complementing the Devils Hole measurements and confirming theoretical equilibrium relationships will allow future researchers to more accurately quantify and describe disequilibrium effects that can affect paleoclimate proxy reconstructions. I recommend acceptance with minor revisions.

Minor suggestions on the manuscript include:

1) Reporting and discussing the difference in pH between Laghetto Basso and Devils Hole. To what extent will this affect fractionation at these sites?

2) Expand on the claim that "the majority of calcites precipitated at the surface achieve neither oxygen-18 nor clumped-isotope equilibrium". You show the Kim and O'Neil experimental data as well as your new foraminifera and bivalve results, but there are other surface calcites with well-constrained crystallization temperatures and $1000\ln\alpha$ values. Consider showing some of these previously-published results or some of previously-calculated equilibrium lines (e.g., Affek and Zaarur, 2014; Tremaine et al., 2011)

3) The sentence on diagenetic and metamorphic carbonates should not be its own paragraph. Either expand on it or include it as part of the previous paragraph.

4) Consider plotting the results of the foraminifera and bivalve d_{18O} measurements in Figure 1. It would complement Figure 2A better.

Minor comments on code:

1) Lines 25-31, 383-386, 413-416: Because you do not cite these publications in the text, please include full citations here.

2) Lines 74-75, 102, throughout: Please either remove commented lines of code, or describe under what circumstances a future researcher may want to un-comment those lines of code.

Reviewer #2 (Remarks to the Author):

This manuscript addresses an issue of wide importance in low-temperature carbonate isotope geochemistry. Although the paper provides no fundamentally new insights it add an important

data point (from an alpine cave in Tuscany) which nicely confirms the assertion by Coples (2007) that extremely slow precipitating subaqueous cave sites may be the best natural examples for "true" isotope equilibrium. As a consequence the vast majority of calcite formed elsewhere near the Earth surface precipitated out of equilibrium, in particular those from biogenic systems. I found the text well written and the figures instructive.

I am not an expert on clumped isotopes but I am aware of interlaboratory differences which may affect the results of this part of the paper. The authors mention this aspect but it is not clear in which direction the data would be shifted if the authors would have followed other protocols. This is a technical aspect and I trust the authors have thought about this carefully.

One minor other point. The authors mention that calcite from both studied sites is characterized by "similar crystalline structures" but it is not clear why is meant here. The two references (31, 32) are not informative about this aspect.

Reviewer #3 (Remarks to the Author):

I have reviewed the manuscript entitled "Most Earth-surface Calcites Precipitate Out of Isotopic Equilibrium" by Daëron et al., submitted for publication in Nat. Commun. This study provides relevant results and insights into the effect of isotope disequilibrium during carbonate precipitation. The authors calculate equilibrium oxygen isotope fractionation factors between water and calcite from subaqueous carbonate speleothems formed at different temperatures by assuming equilibrium carbonate precipitation. These fractionation factors differ significantly from those reported in previous studies based on laboratory experiments and other materials, which are assumed to precipitate under non-equilibrium conditions. Clumped isotopes ($\Delta 47$) are applied to demonstrate that the "equilibrium" $\Delta 47$ -Temperature relationship obtained from subaqueous carbonate speleothems is parallel but offset from that obtained from other carbonate materials (i.e. biogenic carbonate). The manuscript is clear and well-organized. In my opinion, this study deserves to be published in Nat. Commun. However, I found several issues that the authors should address before a new submission.

1. The manuscript's title looks a bit "pretentious" to me. I agree that biogenic carbonates and other carbonates that form under presumed non-equilibrium conditions represent an important percentage of all Earth's carbonate. However, what about other subaqueous carbonate deposits, like authigenic abiogenic carbonates in lakes, abiogenic carbonates in shallow/deep marine environments or hydrothermal and low-temperature carbonate in aquifers? Can those be assumed to precipitate under non-equilibrium conditions? Something is said about it in the "broader application" section, but I miss a deeper discussion. Also, I think the author may find a more accurate title for this study, which main finding is that subaqueous carbonate speleothems seem to better represent isotope equilibrium conditions than biogenic carbonates.

2. In Figure 2A, the X axis is labeled as $1/T^2$, however the units are given in °C, what is not right. The author may use a $1/T^2$ axis on the lower part and a T on the upper part. Same for figure 1. Also, I wonder if other datasets from previous studies could be included in this figure (e.g. the travertine and tufa calibration by Kele et al., 2015 and the recent calibration by Breitenbach et al., 2018, based on cave pearls. They used a similar method for clumped isotopes). Indeed, the Breitenbach et al., 2018 calibration should fall close to the subaqueous speleothems analyzed in this study, so they also are assumed to form in equilibrium conditions. Panel B may also include other datasets for comparison.

3. The number of analysed speleothem samples (assumed to have formed in equilibrium) is rather small ($n=2$). Also, the $\Delta 47$ -Temperature of one of the speleothem samples is indistinguishable (within analytical errors) from that of the biogenic carbonates from previous studies (Peral et al., 2018) (Fig. 2A). This means that the slope of the regression $\Delta 47$ -Temperature line for subaqueous speleothems in this study is controlled by the Devils hole speleothem datapoint. The authors claim that this difference is statistically significant; however, how significant can be a regression line based on two data points? This is important, because all the story of this manuscript revolves around these differences in slopes and offsets. My point is: Is the slope of the two-points line as significant as the one from the biogenic carbonates in Fig. 2A? I wonder if the authors may support their conclusions by using the cave pearl calibration by Breitenbach et al., 2018 or by measuring

more samples from the same/other locations.

4. There is an issue regarding the uncertainties of the calibrations lines that I think may be easily solved by the authors. The analytical errors and instrumental errors of the temperature measurements should be propagated in the equations in page 4, for both the slope and the cut with the y-axis. Also, a 95% confidence level should be included graphically in Figures 1 and 2A. The same approach should be applied to the biogenic carbonate dataset.

5. Because of the $\Delta 47$ -Temperature for subaqueous carbonate speleothems seems to have a different slope compared to biogenic carbonates, the offset between the two type of carbonate samples differs with temperature. Low-temperature biogenic carbonate are similar to low-temperature subaqueous carbonate speleothems; however, at high temperature things seem to be different (Fig. 2A). Can the authors discuss why is this? Can we learn anything from this? Do low-temperature biogenic carbonate precipitate in conditions closer to equilibrium? Can the author guess why is that?

6. The supplementary material supplied by the authors needs more attention. Each table should include a caption with an explanation of its content and the meaning of each column. Otherwise, I do not think these data can be used by others. The number of aliquots for each sample should be indicated, the results of $\delta^{13}\text{C}$ and reproducibility should be provided. The origin of the biogenic carbonates (coordinates, etc.) should be also included in these tables, or at least a clear reference to the original paper (Peral et al., 2018).

7. The author missed a recent study that applies clumped isotopes to subaqueous carbonate speleothems (Gázquez et al., 2018). I think this is an important reference that should be included in the manuscript and discussed/used when necessary, so it deals with similar subaqueous speleothems.

8. In page 3 is said: "In situ observations, spanning 10 years, of pH and temperature, as well as isotopic and elemental analyses of water samples, demonstrate that modern pool water is thermally and chemically stable, with $\text{pH} = 8.2 \pm 0.1$ and $T = 7.9 \pm 0.2 \text{ }^\circ\text{C}$ (1SD). In contrast to most karstic environments of paleoclimatic interest, dripwater must percolate through the Corchia Cave system for long durations on the order of years to decades before reaching Laghetto Basso." Is this 10-years monitoring published? Please, cite it. Also, note that other studies normally deal with dripwater that is probably more variable than a subterranean lake, which volume (and inertia) is larger. The temperature of large water bodies in caves normally varies less than in dripwater in a short-term. So, I don't think the statement "for long durations on the order of years to decades before reaching Laghetto Basso" can be supported by the data, unless you have further evidence for this. It may be that water equilibrates not in the epikarst but in the cave (?).

9. At the end of page 3, you should say that the $\delta^{18}\text{O}$ of the water was also measured in order to calculate fractionation factors. Also, some details about the method used to measure the $\delta^{18}\text{O}$ of waters in this study should be given.

10. I am not sure that the second paragraph in page 5 is very useful and supplies something new for the "broader application". Instead, I am missing in this section a reference to quantitative paleoclimate studies (temperature and $\delta^{18}\text{O}_w$) from subaqueous speleothems (e.g. those analysed in this study and Gázquez et al., 2018).

11. I am not sure the analytical errors of clumped isotopes should be given in 1SE (Table 1 and figures). Fernandez et al., 2018 suggest errors should be reported at specific confidence levels (e.g. 95% CL).

12. The author should note that, in addition to the fractionation factor for slow precipitation of calcite in caves calculated by Coplen, (2007), Tremaine et al., (2011) determined an in-situ fractionation factor for cave carbonates. I think this should be plotted in Fig. 1 and compared to the results of the subaqueous speleothems.

References

T. B. Coplen. Calibration of the calcite–water oxygen-isotope geothermometer at Devils Hole, Nevada, a natural laboratory. *Geochimica et Cosmochimica Acta* 71:16 (2007), pp. 3948–395

S. F. M. Breitenbach, M. J. Mlonek-Vautravers, A.-L. Grauel, L. Lo, S. M. Bernasconi, Inigo A. Müller, James Rolfe, Fernando Gázquez, Mervyn Greaves & D. A. Hodell. Coupled Mg/Ca and clumped isotope analyses of foraminifera provide consistent water temperatures. *Geochimica et Cosmochimica Acta* (2018).

Kele, S., Breitenbach, S.F.M., Capezzuoli, E., Nele Meckler, A., Ziegler, M., Millan, I.M., Kluge, T., Deák, J., Hanselmann, K., John, C.M., Yan, H., Liu, Z., Bernasconi, S.M., 2015. Temperature dependence of oxygen- and clumped isotope fractionation in carbonates: a study of travertines and tufas in the 6–95 °C temperature range. *Geochim. Cosmochim. Acta* 168, 172–192.

Fernando Gázquez, Andrea Columbu, Jo De Waele, Sebastian F.M. Breitenbach, Ci-Rong Huang, Chuan-Chou Shen, Yanbin Lu, José-María Calaforra, Maryline J. Mlonek-Vautravers, David A. Hodell. 2018. Quantification of paleo-aquifer changes using clumped isotopes in subaqueous carbonate speleothems. *Chemical Geology*. 493: 246-257.

M. Peral, M. Daëron, D. Blamart, F. Bassinot, F. Dewilde, N. Smialkowski, G. Isguder, J. Bonnin, F. Jorissen, C. Kissel, E. Michel, N. Vázquez Riveiros & C. Waelbroeck. Updated calibration of the clumped isotope thermometer in planktonic and benthic foraminifera. *Geochimica et Cosmochimica Acta* (2018).

Tremaine, D.M., Froelich, P.N., Wang, Y. 2011. Speleothem calcite formed in situ: Modern calibration of $\delta^{18}\text{O}$ and $\delta^{13}\text{C}$ paleoclimate proxies in a continuously-monitored natural cave system. *Geochimica et Cosmochimica Acta* 75, 4929–4950.

NCOMMS-18-23230
(“Most Earth-surface Calcites Precipitate Out of Isotopic Equilibrium”)
Response to reviewers' comments

Reviewer #1 (Remarks to the Author):

This is a strong paper that adds further evidence that oxygen-isotope equilibrium in calcite is the exception rather than the rule. Complementing the Devils Hole measurements and confirming theoretical equilibrium relationships will allow future researchers to more accurately quantify and describe disequilibrium effects that can affect paleoclimate proxy reconstructions. I recommend acceptance with minor revisions.

Minor suggestions on the manuscript include:

1) Reporting and discussing the difference in pH between Laghetto Basso and Devils Hole. To what extent will this affect fractionation at these sites?

Good point. We now report pH, ionic strength and growth rate in Table 1. We also discuss the possible influence of pH effects (for oxygen-18 and clumped isotopes) in the text.

2) Expand on the claim that "the majority of calcites precipitated at the surface achieve neither oxygen-18 nor clumped-isotope equilibrium". You show the Kim and O'Neil experimental data as well as your new foraminifera and bivalve results, but there are other surface calcites with well-constrained crystallization temperatures and $1000\ln\alpha$ values. Consider showing some of these previously-published results or some of previously-calculated equilibrium lines (e.g., Affek and Zaarur, 2014; Tremaine et al., 2011)

It is well documented that rapid CO₂ degassing results in oxygen-18 disequilibrium between dissolved inorganic carbon (DIC) species and water, increasing $\delta^{18}\text{O}$ in carbonate precipitates (e.g., Hendy, 1971; Mickler et al., 2006; Guo, 2009; Daëron et al. 2011, Kluge & Affek, 2012). It would be quite difficult to claim that the observations reported by Tremaine et al. (2011) and Affek & Zaarur (2014) are not affected by this process.

As we note in the manuscript, calibrating the oxygen-18 thermometer is one of the oldest issues in stable-isotope geochemistry. We have thus updated Figure 1 to include observations on biogenic and synthetic carbonates from two historically important studies, that of Epstein et al. (1953) and of Tarutani et al. (1969).

3) The sentence on diagenetic and metamorphic carbonates should not be its own paragraph. Either expand on it or include it as part of the previous paragraph.

This sentence does indeed belong to the previous paragraph. In the initial submission PDF it appeared to be its own paragraph because of an unfortunate page break, now corrected.

4) Consider plotting the results of the foraminifera and bivalve $\delta^{18}\text{O}$ measurements in Figure 1. It would complement Figure 2A better.

Peral et al. (2018) used the *Kim & O'Neil* (1997) relationship between T and oxygen-18 to constrain their calcification temperatures (note that the clumped-isotope discrepancies between foraminifera and mammillary calcite shown in Figure 2 would only increase if one were to use a different relationship, e.g. that of *Shackleton*, 1974). It would thus be circular logic to include these data in Figure 1.

By contrast, bivalve calcification temperatures were constrained through *in situ* measurements, but they are known to display complex vital effects whose mechanisms are still debated (e.g., *McConnaughey*, 2003, and references therein), which are beyond the scope of this manuscript. A detailed report of oxygen-18 and clumped-isotope disequilibria in various bivalve species, including juvenile and seasonal effects, is about to be submitted for publication (*Huyghe et al.*, in prep.).

Minor comments on code:

1) Lines 25-31, 383-386, 413-416: Because you do not cite these publications in the text, please include full citations here.

Done.

2) Lines 74-75, 102, throughout: Please either remove commented lines of code, or describe under what circumstances a future researcher may want to un-comment those lines of code.

Done.

Reviewer #2 (Remarks to the Author):

This manuscript addresses an issue of wide importance in low-temperature carbonate isotope geochemistry. Although the paper provides no fundamentally new insights it add an important data point (from an alpine cave in Tuscany) which nicely confirms the assertion by *Coplen* (2007) that extremely slow precipitating subaqueous cave sites may be the best natural examples for "true" isotope equilibrium. As a consequence the vast majority of calcite formed elsewhere near the Earth surface precipitated out of equilibrium, in particular those form biogenic systems. I found the text well written and the figures instructive.

I am not an expert on clumped isotopes but I am aware of interlaboratory differences which may affect the results of this part of the paper. The authors mention this aspect but it is not clear in which direction the data would be shifted if the authors would have followed other protocols. This is a technical aspect and I trust the authors have thought about this carefully.

It is indeed quite challenging to predict how results from different laboratories with different standardization protocols would compare. Our claim that the observed T/Δ_{47} relationships in biogenic and slow-growing calcites are statistically different (at 95 % confidence level) is robust because it is based on data obtained in a single lab, using the same carbonate standards, within a relatively short time frame. We are optimistic (but it still remains to be demonstrated) that our use of readily available carbonate standards, already used in a number of laboratories, will allow other groups to compare their results to ours.

One minor other point. The authors mention that calcite from both studied sites is characterized by "similar crystalline structures" but it is not clear why is meant here. The two references (31, 32) are not informative about this aspect.

We changed the wording to make it more specific ("similar surface textures and crystal fabrics").

Reviewer #3 (Remarks to the Author):

1. The manuscript's title looks a bit "pretentious" to me. I agree that biogenic carbonates and other carbonates that form under presumed non-equilibrium conditions represent an important percentage of all Earth's carbonate. However, what about other subaqueous carbonate deposits, like authigenic abiotic carbonates in lakes, abiotic carbonates in shallow/deep marine environments or hydrothermal and low-temperature carbonate in aquifers? Can those be assumed to precipitate under non-equilibrium conditions? Something is said about it in the "broader application" section, but I miss a deeper discussion. Also, I think the author may find a more accurate title for this study, which main finding is that subaqueous carbonate speleothems seem to better represent isotope equilibrium conditions than biogenic carbonates.

This comment by Reviewer #3 made us aware that the original manuscript failed to make it clear that the critical factor linking Devils Hole and Laghetto Basso calcites is not that they are subaqueous, but rather their extremely slow crystallization rates. We tried to clarify this point in the revised manuscript and we believe that the new version is significantly improved in that regard.

Regarding the title, we were careful to qualify it ("Most Earth-surface Calcites" being much more restrictive than "All Calcites"). Authigenic lake carbonates appear to precipitate rapidly, with potentially large kinetic oxygen-18 fractionations (*Fronval et al.*, 1995). We do mention hydrothermal carbonates and deep marine carbonates in the text as potential exceptions but we stand by our original title, which we believe to be forceful rather than pretentious.

2. In Figure 2A, the X axis is labeled as $1/T^2$, however the units are given in °C, what is not right. The author may use a $1/T^2$ axis on the lower part and a T on the upper part.

Done.

Same for figure 1.

Done.

Also, I wonder if other datasets from previous studies could be included in this figure (e.g. the travertine and tufa calibration by *Kele et al.*, 2015 and the recent calibration by *Breitenbach et al.*, 2018, based on cave pearls. They used a similar method for clumped isotopes). Indeed, the *Breitenbach et al.*, 2018 calibration should fall close to the subaqueous speleothems analyzed in this study, so they also are assumed to form in equilibrium conditions. Panel B may also include other datasets for comparison.

As mentioned above and in the manuscript, we cannot yet compare results between different laboratories with the necessary precision (~10 ppm), which is why our dataset presents a rare opportunity to compare biogenic and slow-growing carbonates. By way of illustration, despite the use of identical standards (but different analytical protocols), the modern foraminifer observations reported by *Breitenbach et al.* (2018) yield Δ_{47} values 10–20 ppm lower on average than those reported by *Peral et al.* (2018).

Beyond these inter-lab standardization issues, we don't believe that a good case can be made that travertines, tufas, and cave pearls truly achieve isotopic equilibrium. The growth rates quoted by *Kele et al.* (2015) correspond to $\log(R)$ values between -7.2 and -4.5, which places these carbonates close to the kinetic limit (see for example figure 6 from *Watkins & Hunt*, 2015). The cave pearls analyzed by *Breitenbach et al.* (2018) have poorly constrained growth rates (pers. comm. from S. Breitenbach), but published cave pearl growth rates are quite rapid, between 0.2 and 1 mm/yr (*McKinstry*, 1931; *Donahue*, 1965; *Kirchmayer* 1987; *Melim & Spilde*, 2011), three orders of magnitude faster than our slow-growing samples.

In both cases, these rapid growth rates are primarily driven by CO₂ degassing. Although both *Kele et al.* (2015) and *Breitenbach et al.* (2018) state that their carbonate samples grow under conditions that minimize degassing, they do not exclude prior degassing (in the case of cave pearls, pool calcite/aragonite and tufa) nor partial but rapid degassing (in the case of hydrothermal vents), both of which are likely to produce kinetic oxygen-18 enrichments in carbonate. One cannot claim that these samples form in isotopic equilibrium based only on their relatively high $\delta^{18}\text{O}$ values, which are fairly typical of speleothems as a whole (e.g., figure 7 from *Tremaine et al.*, 2011), and these degassing effects are also known to result in large Δ_{47} offsets (*Guo*, 2009; *Daëron et al.*, 2011; *Kluge & Affek*, 2012).

3. The number of analysed speleothem samples (assumed to have formed in equilibrium) is rather small (n=2). Also, the Δ_{47} -Temperature of one of the speleothem samples is indistinguishable (within analytical errors) from that of the biogenic carbonates from pervious studies (*Peral et al.*, 2018) (Fig. 2A). This means that the slope of the regression Δ_{47} -Temperature line for subaqueous speleothems in this study is controlled by the Devils hole speleothem datapoint. The authors claim that this difference is statistically significant; however, how significant can be a regression line based on two data points? This is important, because all the story of this manuscript revolve around these differences in slopes and offsets. My point is: Is the slope of the two-points line as significant as the one from the biogenic carbonates in Fig. 2A?

There are two different issues at stake in this comment: whether the observed slope differences are statistically significant (i.e. unlikely to result from measurement errors alone), and whether the samples used to constrain these slopes provide good (unbiased) estimates of, for instance, calcites in isotopic equilibrium. We have high confidence in both of these assertions.

We would obviously prefer to have access to a larger number of very slow-growing calcites of well-known formation temperature, but such samples remain extremely rare. We would argue that a line defined by two points, 25 °C apart, is a major improvement relative to the earlier *status quo* of an equilibrium line constrained by a single observation. What's more, the fact that our observations match almost perfectly with the theoretical prediction of *Watkins et al.* (2014) supports our assumption that these two samples provide reliable constraints on isotopic equilibrium.

I wonder if the authors may support their conclusions by using the cave pearl calibration by *Breitenbach et al.*, 2018

We note that, to the best of our knowledge, these specific cave pearls have only been very briefly described by *Breitenbach et al.* (2018) in the context of a report focussing on foraminiferal records. More generally, see our comments above regarding (a) the assumption that cave pearls record isotopic equilibrium, and (b) the difficulty of precisely comparing clumped-isotope measurements from the same lab.

or by measuring more samples from the same/other locations.

As stated above, we would be delighted to do so, but unfortunately such samples are extremely rare. In the ten years elapsed since the original findings of *Coplen* (2007), to the best of our knowledge no other equivalent calcites had been reported so far.

4. There is an issue regarding the uncertainties of the calibrations lines that I think may be easily solved by the authors. The analytical errors and instrumental errors of the temperature measurements should be propagated in the equations in page 4, for both the slope and the cut with the y-axis.

Done.

Also, a 95% confidence level should be included graphically in Figures 1 and 2A. The same approach should be applied to the biogenic carbonate dataset.

In Figure 1, we added the 95 % confidence region for Equation (1). Figure 2A, however, is already quite busy as it is, and try as we might, adding confidence regions for the regressions makes it illegible. We believe that information regarding the uncertainties on the three regressions is best displayed in Figure 2B, which we have improved by displaying the probability distributions for each regression slope.

5. Because of the $\Delta 47$ -Temperature for subaqueous carbonate speleothems seems to have a different slope compared to biogenic carbonates, the offset between the two type of carbonate samples differs with temperature. Low-temperature biogenic carbonate are similar to low-temperature subaqueous carbonate speleothems; however, at high temperature things seem to be different (Fig. 2A). Can the authors discuss why is this? Can we learn anything from this? Do low-temperature biogenic carbonate precipitate in conditions closer to equilibrium? Can the author guess why is that?

Although one can always offer “intuitive” guesses as to why the three regressions lines converge at low temperatures, in our experience there is limited value in doing so, because many competing factors (e.g., saturation state, crystallization kinetics, metabolic effects, most kinetic isotope fractionation factors, DIC equilibration kinetics...) vary with temperature. We believe such issues are best addressed through comprehensive isotopic models such as those, among others, of *Watkins & Hunt (2015)* or *Devriendt et al. (2017)*. This is clearly beyond the scope of our manuscript, and here we voluntarily refrain from conjecture.

6. The supplementary material supplied by the authors needs more attention. Each table should include a caption with an explanation of its content and the meaning of each column. Otherwise, I do not think these data can be used by others.

We note that the supplementary materials provided with our original submission only included (1) Python code designed to process raw analytical data, perform statistical computations, and generate the tables and figures included in the manuscript; (2) the corresponding raw analytical data in CSV format; (3) the foraminifer dataset reported by *Peral et al. (2018)*. The intent of providing this code is to make our results fully reproducible by anyone. In light of Reviewer #3's concerns, we added proper documentation in the form of a “readme” document, which is standard practice for scientific code of this kind.

The number of aliquots for each sample should be indicated; the results of $d_{13}C$ and reproducibility should be provided.

We added this information to the methods section and in Table S2.

The origin of the biogenic carbonates (coordinates, etc.) should be also included in these tables, or at least a clear reference to the original paper (*Peral et al., 2018*).

The bivalve samples are described in detail (including coordinates) in the Methods section. We added a short description of the foraminifer samples to the Methods section, including a reference to the original article.

7. The author missed a recent study that applies clumped isotopes to subaqueous carbonate speleothems (Gázquez et al., 2018). I think this is an important reference that should be included in the manuscript and discussed/used when necessary, so it deals with similar subaqueous speleothems.

As mentioned above, Devils Hole and Laghetto Basso calcites do not achieve isotopic equilibrium simply because they grow under water, but rather because they crystallize extremely slowly.

Gázquez et al. (2018) report U/Th, stable isotope and clumped isotope measurements performed on aragonite/calcite flowstones, dogtooth and scalenohedral calcite spars and calcite coatings whose growth rates remain essentially unknown. They assume that their subaqueous samples formed in isotopic equilibrium, but offer no strong argument for doing so. We note that some of those samples, such as flowstones, which form under conditions favorable to rapid degassing, are very unlikely to have formed at isotopic equilibrium. Only the “cave cloud” precipitates of Santa Barbara 2 resemble our mammillary calcite coatings, but according to Table 1 and Figure 6 from Gázquez et al. (2018), they most likely formed between ~400 ka and 247 ± 17 ka. These age constraints, combined with the coating thickness of 60–80 cm reported by De Waele & Forti (2006), yield a very rough estimate of 4-5 $\mu\text{m}/\text{yr}$ which is still 10 times faster than at Devils Hole and Laghetto Basso.

For the above reasons, we do not share Gázquez et al.'s confidence that their samples truly record isotopic equilibrium, but this does not greatly affect their main findings, which are based on age constraints and large temperature differences (from 13 to 130 °C). We believe it would not be constructive to point out that we disagree with the assumptions made in this particular study.

8. In page 3 is said: “In situ observations, spanning 10 years, of pH and temperature, as well as isotopic and elemental analyses of water samples, demonstrate that modern pool water is thermally and chemically stable, with $\text{pH} = 8.2 \pm 0.1$ and $T = 7.9 \pm 0.2$ °C (1SD). In contrast to most karstic environments of paleoclimatic interest, dripwater must percolate through the Corchia Cave system for long durations on the order of years to decades before reaching Laghetto Basso.” Is this 10-years monitoring published? Please, cite it.

Done.

Also, note that other studies normally deal with dripwater that is probably more variable than a subterranean lake, which volume (and inertia) is larger. The temperature of large water bodies in caves normally varies less than in dripwater in a short-term. So, I don't think the statement “for long durations on the order of years to decades before reaching Laghetto Basso” can be supported by the data, unless you have further evidence for this.

This statement is based on the conclusions of Piccini et al. (2008), based on tritium measurements. We added the relevant reference to the text.

It may be that water equilibrates not in the epikarst but in the cave (?).

The epikarst above Corchia Cave is very shallow and the cave system very large. We make no mention of the epikarst in the text. It is indeed very likely that water dripping into Laghetto Basso has had ample time, while percolating through the cave system, to reach isotopic equilibrium. We believe this point is made clear in the manuscript.

9. At the end of page 3, you should say that the $\text{d}18\text{O}$ of the water was also measured in order to calculate fractionation factors. Also, some details about the method used to measure the $\text{d}18\text{O}$ of waters in this study should be given.

Good point. We now explicitly mention the isotopic composition of Devils Hole and Laghetto Basso waters, with references to the original publications.

10. I am not sure that the second paragraph in page 5 is very useful and supplies something new for the "broader application".

Although we do not share Reviewer #3's opinion regarding the usefulness of this passage, we recognize that it was not a good choice of opening paragraph for the "broader implications" section, and we moved it to the end of the preceding section.

Instead, I am missing in this section a reference to quantitative paleoclimate studies (temperature and $\delta^{18}\text{O}$) from subaqueous speleothems (e.g. those analysed in this study and Gázquez et al., 2018).

See above our response regarding Gázquez et al. (2018).

More generally, we acknowledge that many published studies assume that carbonates form close to isotopic equilibrium, and our results imply that this is not the default situation for Earth-surface carbonates. However we believe it would be neither fair nor constructive to criticize specific past studies. We have thus rewritten part of this section to make our point even more clearly, without picking on specific past studies.

11. I am not sure the analytical errors of clumped isotopes should be given in 1SE (Table 1 and figures). Fernandez et al., 2018 suggest errors should be reported at specific confidence levels (e.g. 95% CL).

We must respectfully disagree. Fernandez et al. warned against the common practice of assigning standard errors based on a small number of observations, and argued that in such cases one should at least report 95% confidence levels, as it is well-known that this practice yields non-Gaussian estimators of the mean. In this manuscript, however, we instead use a much more robust estimate of standard errors, based on a large number ($N = 244$) of standard and sample measurements (for more details see section 3.1 of Peral et al., 2018). With such a large number of observations, Student's t -distribution statistics become indistinguishable from a Gaussian behavior, so that the 95 % confidence limits on our Δ_{47} values are extremely close to ± 1.96 times the standard errors quoted in Table 1, as expected in the Gaussian case.

12. The author should note that, in addition to the fractionation factor for slow precipitation of calcite in caves calculated by Coplen, (2007), Tremaine et al., (2011) determined an in-situ fractionation factor for cave carbonates. I think this should be plotted in Fig. 1 and compared to the results of the subaqueous speleothems.

Tremaine et al. (2011) reported oxygen-18 fractionation factors for speleothems from a number of cave sites. As mentioned above, most speleothems precipitate from DIC solutions affected by kinetic effects associated with CO_2 degassing, resulting in oxygen-18-enriched precipitates (e.g., Hendy, 1971; Daëron et al., 2011; Kluge & Affek, 2012). We added a sentence regarding this issue in the second paragraph of our "broader implications" section.

References

- Affek & Zaarur (2014)
Kinetic isotope effect in CO₂ degassing: Insight from clumped and oxygen isotopes in laboratory precipitation experiments, *Geochimica et Cosmochimica Acta* 143.
- Daëron et al. (2011)
¹³C¹⁸O clumping in speleothems: Observations from natural caves and precipitation experiments, *Geochimica et Cosmochimica Acta* 75.
- Davidson & McKinsty (1931)
"Cave pearls", oolites, and isolated inclusions in veins, *Economic Geology* 26, 289–294.
- De Waele & Forti (2006).
A new hypogean karst form: the oxidation vent, *Z. Geomorphol.* 147, 107–127.
- Donahue (1965)
Laboratory growth of pisolite grains, *Journal of Sedimentary Petrology* 39, 251–256.
- Fronval et al. (1995)
Oxygen isotope disequilibrium precipitation of calcite in Lake Arresø, Denmark, *Geology* 23:5, 463–466.
- Guo (2009)
Carbonate clumped isotope thermometry: application to carbonaceous chondrites and effects of kinetic isotope fractionation. PhD dissertation. California Institute of Technology.
- Hendy (1971)
The isotopic geochemistry of speleothems: I. The calculation of the effects of different modes of formation on the isotopic composition of speleothems and their applicability as palaeoclimatic indicators. *Geochimica et Cosmochimica Acta* 35.
- Kim & O'Neil (1997)
Equilibrium and nonequilibrium oxygen isotope effects in synthetic carbonates, *Geochimica et Cosmochimica Acta* 61.
- Kirchmayer (1969)
Kristallisations- und Rekristallisationsgefüge in Höhlenperlen aus Bergwerken, *Osterreichische Akademie der Wissenschaften, Sitzungsberichte, Mathematisch-Naturwissenschaftliche Klasse, Abteilung I: Biologie, Mineralogie, Erdkunde und Verwandte Wissenschaften*, v. 177, p. 233–245.
- Kluge & Affek (2012)
Quantifying kinetic fractionation in Bunker Cave speleothems using Δ_{47} , *Quaternary Science Reviews* 49.
- McConnaughey, 2003
Sub-equilibrium oxygen-18 and carbon-13 levels in biological carbonates: carbonate and kinetic models, *Coral Reefs* 22.
- Mickler et al. (2006)
Large kinetic isotope effects in modern speleothems, *Geological Society of America Bulletin* 118.
- Peral et al. (2018)
Updated calibration of the clumped isotope thermometer in planktonic and benthic foraminifera, *Geochimica et Cosmochimica Acta* 239.
- Tremaine et al. (2011)
Speleothem calcite formed *in situ*: Modern calibration of $\delta^{18}\text{O}$ and $\delta^{13}\text{C}$ paleoclimate proxies in a continuously-monitored natural cave system, *Geochimica et Cosmochimica Acta* 75.

Reviewer #3 (Remarks to the Author):

I have reviewed the revised version of the manuscript entitled "Most Earth-surface Calcites Precipitate Out of Isotopic Equilibrium" by Daëron et al.

The authors have provided reasonable answers to most of the comments made by the reviewers. In my opinion the manuscript is ready for publication and I would be delighted to see it published in Nat. Comm.

Cheers